# Beyond Destabilizing Activity of SAP11-like Effector of *Candidatus* Phytoplasma mali Strain PM19

**DOI:** 10.3390/microorganisms10071406

**Published:** 2022-07-12

**Authors:** Kajohn Boonrod, Alisa Strohmayer, Timothy Schwarz, Mario Braun, Tristan Tropf, Gabi Krczal

**Affiliations:** RLP AgroScience GmbH, Breitenweg 71, 67435 Neustadt, Germany; alisa.strohmayer@web.de (A.S.); timothy.schwarz@agrosciecne.rlp.de (T.S.); mario.braun@agroscience.rlp.de (M.B.); tristantropf@web.de (T.T.); gabi.krczal@agroscience.rlp.de (G.K.)

**Keywords:** phytoplasma, SAP11, TCP, *Candidatus* Phytoplasma mali, destabilization

## Abstract

It was shown that the SAP11 effector of different *Candidatus* Phytoplasma can destabilize some TEOSINE BRANCHES/CYCLOIDEA/PROLIFERATING CELL FACTORs (TCPs), resulting in plant phenotypes such as witches’ broom and crinkled leaves. Some SAP11 exclusively localize in the nucleus, while the others localize in the cytoplasm and the nucleus. The SAP11-like effector of *Candidatus* Phytoplasma mali strain PM19 (SAP11_PM19_) localizes in both compartments of plant cells. We show here that SAP11_PM19_ can destabilize TCPs in both the nucleus and the cytoplasm. However, expression of SAP11_PM19_ exclusively in the nucleus resulted in the disappearance of leaf phenotypes while still showing the witches’ broom phenotype. Moreover, we show that SAP11_PM19_ can not only destabilize TCPs but also relocalizes these proteins in the nucleus. Interestingly, three different transgenic Nicotiana species expressing SAP11_PM19_ show all the same witches’ broom phenotype but different leaf phenotypes. A possible mechanism of SAP11-TCP interaction is discussed.

## 1. Introduction

Phytoplasmas are plant pathogenic bacteria that are transmitted by insect vectors and reside in the phloem of their plant hosts. Phytoplasmas secrete effector proteins that change plant development and increase phytoplasma fitness [1,2]. One of these secreted proteins of Aster Yellows phytoplasma strain Witches’ Broom (AY-WB) is SAP11 (SAP11_AY-WB_), a small effector protein that has been extensively investigated. SAP11_AY-WB_ specifically localizes in the plant cell nucleus via a nuclear localization sequence (NLS) within the protein and plant importin α [3], while SAP11 of *Cadidatus* Phytoplasma mali (SAP_mali_) strain PM19 (SAP11_PM19_) localizes in the cytoplasm and nucleus [4,5]. Transgenic *A. thaliana* lines expressing SAP11 show severe symptoms, including crinkled leaves, crinkled siliques, stunted growth and an increase in stem number and the witches’ broom phenotype [1,4]. Analysis of transgenic plants expressing SAP11 shows that SAP11 binds and destabilizes CINCINNATA (CIN)-related TEOSINTE BRANCHED1, CYCLOIDEA, and PROLIFERATING CELL FACTORS transcription factors (TCPs), resulting in a decrease in jasmonate (JA) production [6], an alteration of volatile organic compounds (VOC) production [7] and an increase in insect vector reproduction [1].

TCPs regulate a variety of plant processes, from plant development to defense responses. The functions of different AtTCPs and their role in biosynthetic processes have been reviewed in detail by Shutian Li [8]. It was shown that the downregulation of AtTCP 3, 4, 10 that are regulated by the microRNA miR319 leads to crinkled leaves and/or siliques phenotypes [9,10]. Therefore, the crinkled leaf phenotype in the transgenic plants expressing SAP11 are probably due to the destabilization of these TCPs in planta. Some bacterial effectors can bind some TCPs and relocated these proteins into the nucleus. HopBB1, a *Pseudomonas syringae* Type III effector, binds AtTCP14 and JAZ3 and relocalizes the complex in the nucleus, resulting in the degradation of the proteins [11]. Moreover, PRR2 protein, a pseudo-response regulator, interacts with AtTCP19 or AtTC20 in planta and relocates the proteins into cajal bodies or in nuclear speckles, respectively [12]. However, the relocalization of TCPs by SAP11 has not been reported yet. 

Despite differences in the amino acid sequences of AY-WB_SAP11 and AP_SAP11-like protein, it was shown that the SAP11-like proteins of ‘*Ca*. P. mali’ strain AT (Apfeltribsucht) and strain STAA (South Tyrol/Alto Adige) bind and destabilize some TCP transcription factors [5,6]. Transgenic Arabidopsis plants expressing SAP11_PM19_ showed typical phenotypes similar to those expressing AY-WB_SAP11 [1,4]. The similarities of TCP-binding, the biochemical changes in transgenic plants and hydrophobic amino acid patterns, leads to the assumption that these proteins may have a similar function during phytoplasma infection [6].

Strohmayer and co-worker [4] showed that SAP11_PM19_ localizes in the cytoplasm and the nucleus and can destabilize some AtTCPs of class I and II. It is not yet clear in which compartment of plant cells the destabilization of AtTCPs takes place. Moreover, SAP11_PM19_ can bind AtTCP6 and 19 (class I), but it only causes the destabilization of AtTCP6. To date the function of TCP6 is not known, whereasTCP19 is shown to be among the three TCPs (13, 14 and 19) identified as immune interactors. The *tcp19* single mutant plants exhibit enhanced disease susceptibility to two different avirulent *Hyaloperonospora arabidopsidis* isolates, indicating that TCP19 is required for a full immune system function [13]. Therefore, we question what effects SAP11_PM19_ might have on AtTCP19, other than binding. In addition, the effects of SAP11 have been shown to differ in different plant species, likely due to different TCP analogs [14]. To explore the possible additional function of SAP11_PM19_ on AtTCP, especially class I (AtTCP6 and 19), and on TCPs of various Nicotiana species, we further investigated the function of this SAP11_PM19_ in vivo and in vitro.

## 2. Materials and Methods

### 2.1. Origin of SAP11_PM19_ DNA

Due to this, *SAP11_PM19_* (GenBank Accession number MK966431) contains a sequence-variable mosaic protein signal sequence (*SVM*), which is presumed to encode a signal peptide with an unidentified function; therefore, SAP11_PM19_ was amplified from field-collected *Cacopsylla picta* without *SVM* as described by Strohmayer et al. [4]. 

### 2.2. Protein Localization

For localization of SAP11_PM19_, the gene was fused to *GFP* at C termini in pPZP200 under control of *Cauliflower mosaic virus* 35S promotor as described by Strohmayer et al. [4]. To localize the protein exclusively in the cytoplasm or nucleus, the gene was fused to *NES* and *bi-NLS* of DNA rearrangement methyltransferase (DRM) [15] at N termini and with *green fluorescence protein* (*GFP*) gene at C termini resulting in pPZP2000-*NES*-*SAP11_PM19_*-*GFP* and pPZP2000-*NES-SAP11_PM19_*-*GFP*, respectively. To mark the plant nucleus, *red fluorescence protein* (*RFP*) gene was fused to *bi-NLS* of *DRM* at N termini, resulting in pPZP2000-*bi-NLS–RFP*. The plasmids were transformed into bacteria *A. tumefecine*, ATHV strain used for transient expression via agroinfiltration. The localization of the expressed proteins was visualized under a confocal microscope. 

### 2.3. Transient Protein Expression in Planta 

The *SAP11_PM19_* genes was codon optimized for plant expression and synthesized (GeneCust, Ellange, Luxembourg). This version of the gene was used as the basis for all constructs used in in planta experiments in this work. Agroinfiltration was performed as described [16].

### 2.4. Generation of Transgenic A. thaliana and Nicotiana spp. Lines

For producing transgenic *A. thaliana* (Columbia ecotype, col-0) plant lines, a floral dip was performed as previously described [4,17]. F1 and F2 generation of transgenic plants were selected by spraying of BASTA solution, diluted 1/1000 in H_2_O. The F2 generation of transgenic *A. thaliana* lines was screened for phenotypic symptoms and analyzed using RT-qPCR. For establishing transgenic Nicotiana plant lines, the plasmids were transformed into *Agrobacterium tumefaciens* (*A. tumefaciens*) ATHV by electroporation. The bacterial cell suspension was used for leaf disc transformation of *N. benthamiana, N. tabacum* and *N. occidentalis* plants. Transgenic Nicotiana plants were generated and maintained as described by Horsch et al. [18].

### 2.5. RT-qPCR 

For RT-qPCR, RNA was extracted from transgenic *A. thaliana* plants and tested for DNA contamination by performing RT-qPCR with 10 ng and 50 ng RNA per reaction as described by Strohmayer et al. [4]. cDNA was synthesized and subjected to RT-qPCR with four technical replicates as described by Strohmayer et al. [4]. *Glyceraldehyde-3-phosphate dehydrogenase* (*GAPDH*) and *protein phosphatase 2* (*PP2A*) were used *as* standard reference genes due to their stable expression, especially during different developmental stages of *A. thaliana* [19]. The gene-specific primers that can bind to all three variants were used [4]. The relative expression levels were calculated using the ddCt method [20] and normalized to the geometric average of the Cq of the reference gene as described by Strohmayer et al. [4]. 

### 2.6. In Vivo Protein Interaction Assay

To identify protein–protein interaction in planta, we used the BiFC system described by Grefen and Blatt [21]. The binary plasmids for BiFCt-2in1 system were obtained from Addgene. *SAP11_PM19_* and *AtTCP6* or *AtTCP19* were cloned at C-termini of YFP each half in BiFCt-2in1-NN vector, resulting in the plasmid BiFCt-2in1-NN-*SAP11_PM19_*/*AtTCP6* and BiFCt-2in1-*NN-SAP11_PM19_*/*AtTCP19*, respectively. For comparing the localization of AtTCPs in question, *AtTCP6* and *AtTCP19* were cloned in binary pPZP200 vector [4] fused with RFP, resulting in pPZP200-*AtTCP6*/*19-RFP*. The plasmids were then transformed into *A. tumefecine*, ATHV strain for transient protein expression in *N. benthamiana* plant leaves. The protein–protein interaction was visualized under a confocal microscope with GFP filter while RFP filter was used for visualizing the localization of AtTCP-RFP. 

### 2.7. Protein Expression and Purification

*AtTCP19* fused to *hexa Histidine* (*His*) tag and *SAP11_PM19_* were cloned into pMalX2c vector (New England Biolab, Frankfurt, Germany) fused with maltose-binding protein gene (*MBP*) for recombinant protein expression in *E. coli* (BL21^+^). The recombinant proteins were expressed and purified as described by Strohmayer et al. [4]. 

### 2.8. Electro Mobility Shift Assay (EMSA)

The DNA-binding activity of AtTCP was examined as described by Viola et al. [22] with some modifications. Shortly, 1 µg of recombinant MBP-AtTCP19-His was incubated with C6-oligomer [22] fused with Cyanine5 fluorophore (C6-oligo-Cy5). To investigate the effect of SAP11_PM19_ on the DNA-binding activity of MBP-AtTCP19-His, 1 µg of MBP-SAP11_PM19_ was incubated with MBP-AtTCP19-His for 30 min period adding C6-oligo-Cy5. The reactions were electrophoresed in 1.5% TBE agarose gel. The band shift of C6-oligo-Cy5 was visualized using a phosphoimager (Biorad, Germany). 

### 2.9. ELISA

A total of 1 µg of recombinant MBP-AtTCP19-His in PBS was coated in an ELISA plate. After blocking the coated plate with 3% BSA, 1 µg of recombinant MBP-SAP11_PM19_ was added. The binding of MBP-AtTCP19-His and MBP-SAP11_PM19_ was detected using anti-SAP11_PM19_ developed in mouse (David, Germany) followed by anti-mouse-POD (Merck). The substrate was added and the developed color was measured with an ELISA reader at OD450 nm. 

## 3. Result

### 3.1. Transient Expression of SAP11fused GFP in Cytoplasm and Nucleus

SAP11_PM19_ is localized in the nucleus and cytoplasm, and the expression of SAP11_PM19_ in transgenic Arabidopsis plants causes changes in the plant phenotypes [4]. To study the relationship between symptoms and the localization of SAP11_PM19_, three constructs (*SAP11_PM19_*, *SAP11_PM19_* fused to a nuclear export signal (*NES*) of HIV-Rev [23], *NES-SAP11_PM19_* to localize the protein exclusively in the cytoplasm and SAP11_PM19_ fused to the bipartite nuclear leader sequence (*bi-NLS*) of the *Nicotiana tabacum* (*N. tabacum*) domains rearranged methyltransferase 1 [15], *bi-NLS-SAP11_PM19_* to localize the protein into the plant nucleus) were generated. The genes were fused with *GFP* and transiently expressed for protein localization analysis via agroinfiltration and were later used (without GFP) for establishing transgenic *Arabidopsis* plant lines. The results of the transient expression in *N. benthamiana* show that (Figure 1) the wt SAP11_PM19_ is localized in both the cytoplasm and nucleus (Figure 1, middle panel). The protein is predominantly found in the cytoplasm when fused with NES (Figure 1, upper panel) while it is exclusively localized in the nucleus when fused with bi-NLS (Figure 1, lower panel). Thus, these constructs (without GFP) were used for establishing transgenic Arabidopsis plants for studying the effect of the protein localization on the plant phenotype. 

### 3.2. Transgenic Arabidopsis Plants Expressing SAP11_PM19_ and Its Derivatives

Transgenic *A. thaliana* plants expressing NES-SAP11_PM19_ show the typical phenotype (witches’ broom and crinkled leaves), similar to *A. thaliana* plants expressing wt SAP11_PM19_, whereas leaf symptoms disappeared in the transgenic *A. thaliana* plants expressing bi-NLS-SAP11_PM19_ (Figure 2a). However, the number of stems of all transgenic *A. thaliana* plants are comparable (Figure 2b). Thus, the results suggest that the localization of SAP11_PM19_ affect the symptom development of the transgenic plants. Due to the different expression level of the transgenes, which could affect the phenotype development of the transgenic plants, we further analyzed the expression levels of the transgenes using RT-PCR. The results show that the expression level of bi-NLS- SAP11_PM19_ is lower than NES and wt SAP11_PM19_ in all tested plant lines. However, the relative expression levels, means of the expression level normalized in comparison the reference genes, of plant line #5 of wt, line #12 of NES and line #8 of bi-NLS-SAP11_PM19_ are not significantly different (Figure 2c), whereas the symptom development with respect to the leaf morphology of these plants is different, as shown in Figure 2a. Therefore, these results suggest that the relative expression levels of the different transgenes are not the main cause of the difference in leaf phenotype development. 

### 3.3. SAP11_PM19_ Can Destabilizes AtTCPs in Both Cytoplasm and Nucleus

The effector SAP11 can destabilize some AtTCPs, which consequently causes changes in the plant phenotypes such as the development of witches’ broom [1,4,24]. The results in Figure 2 show that when SAP11_PM19_ is expressed exclusively in the nucleus, the crinkled leaf phenotype of transgenic plants disappear. Since SAP11_PM19_ localizes in the cytoplasm and nucleus [4,24], it is not clear in which compartment of the plant the destabilization takes place. Therefore, we question whether the destabilization depends on the localization of SAP11_PM19_. To elucidate this question, the three gene constructs visualized in Figure 1 were transiently co-expressed with AtTCP3 (class II, involved in leaf morphology, [9,10]) and AtTCP6 (class I, bound and destabilized by SAP11_PM19_ [4]) fused to the HA tag for detecting protein expression as described by Strohmayer et al. [4] and the expressed proteins were analyzed in a Western blot. The results in Figure 3 show that SAP11_PM19_ can destabilize AtTCP3-HA and AtTCP6-HA in both compartments, suggesting that the disappearance of the crinkled leaves phenotype in the transgenic plant expressing SAP11_PM19_ exclusively in the nucleus (bi-NLS-SAP11_PM19_) should not be due to the exclusive activity of this SAP11_PM19_ in the nucleus. 

### 3.4. SAP11_PM19_ Cannot Inhibit the DNA-Binding Activity of AtTCP19

It was already shown that SAP11_PM19_ binds but cannot destabilize AtTCP19 [4], therefore we further investigated other possible activities of SAP11_PM19_ on AtTCP19. It was shown that AtTCPs bind to a specific DNA sequence in the nucleus [22]. Therefore, we questioned whether the binding of SAP11_PM19_ to AtTCP19 could inhibit its DNA binding activity. To address this question, we analyzed the DNA binding activity of AtTCP19 in the presence of SAP11_PM19_ using an EMSA. For this purpose, we expressed SAP11_PM19_ and AtTCP19 recombinantly as fusion protein with maltose-binding protein (MBP) for increasing protein solubility in *E. coli* and a hexa-histidine (His) for purification (Figure 4a). The binding activity of recombinant MBP-SAP11_PM19_ to MBP-AtTCP19-His was confirmed in an ELISA assay (Figure 4b). The EMSA result (Figure 4c) shows that MBP-AtTCP19-His can bind to the C6-oligo-Cy5, causing a mobility shift, and the MBP-SAP11_PM19_ cannot inhibit the DNA binding activity of MBP-AtTCP19-His. 

### 3.5. SAP11_PM19_ Relocalizes AtTCP6 and AtTCP19 in the Nucleus

Some bacterial effectors can relocalize TCPs in plant cells [11,12]. To investigate whether SAP11_PM19_ can relocalize AtTCPs in the nucleus, we analyzed this possibility using the BiFC system. AtTCP6 and 19 were selected for this analysis because SAP11_PM19_ binds and destabilizes AtTCP6, whereas it binds but cannot destabilize AtTCP19 [4]. In the BiFC system, *SAP11_PM19_* and *AtTCP19* were fused with each half of *YFP* in the same plasmid and the fused proteins were transiently expressed in *N. benthamiana* via agroinfiltration. The results in Figure 5 show that SAP11_PM19_ can indeed bind AtTCP6 and AtTCP19 and relocalizes AtTCP6 from the nucleolus (Figure 5 upper-left panel) to the nucleoplasm (Figure 5 upper-right panel) and relocalizes AtTCP19 from nucleoplasm (Figure 5 lower-left panel) into nuclear bodies (cajal body, Figure 5 lower-right panel). We further confirmed these results by transiently expressing *AtTCP6* and *19* fused with *RFP* in wt and transgenic plants expressing SAP11_PM19_. The results in Figure 6 confirm that the transiently expressed AtTCP6 and AtTCP19-RFP are relocalized by SAP11_PM19_ in the transgenic plant expressing SAP11_PM19_ in the same manner as shown by BiFC analysis. 

### 3.6. Different Expressing Phenotypes of Transgenic Tobacco Plants Expressing SAP11_PM19_

The symptoms that occur in plants infected with phytoplasmas depend strongly on the type of host plant and the corresponding protein that interacts with the effector, as well as on the type of effector of the phytoplasma. It was shown that SAP11_AY-WB_ and SAP11 of Maize Bushy Stunt Phytoplasma (SAP11_MBSP_) have evolved to target overlapping but distinct class II TCPs of their plant hosts and that these transcription factors also have overlapping but distinct roles in regulating the development in dicot (*A. thaliana*) and monocot (Maize) host plants [14]. We wondered if transgenic plants of the same genus (Nicotiana) but different species (*N. benthamiana*, *N. occientalis* and *N. tabaccum*) would show different phenotypes when expressing SAP11_PM19_. We therefore established transgenic plants of three different tobacco species expressing SAP11_PM19_. The phenotypes of these transgenic tobacco plants were compared with transgenic *A. thaliana* expressing SAP11_PM19_ (Figure 2). The results in Figure 7 show that all transgenic tobacco species expressing SAP11_PM19_ exhibit the witches’ broom phenotype, while the crinkled leaf phenotype is only found in transgenic *N. occidentalis* plants. However, the leave of the other two transgenic tobacco species showed a rough leaf surface. Although studying the interaction of SAP11_PM19_ with TCPs of the different tobacco species are difficult due to limited genomic data, our results strongly suggest that the use of a model plant to study the SAP11 effector does not necessarily reflect the fully typical symptoms of the natural phytoplasma-infected host plant in question.

## 4. Discussion

SAP11 is an effector found in different species of phytoplasmas. Although the protein sequences differ between phytoplasma species, these effectors share the same activity in binding and destabilizing TCPs [1,4,5,6,24]. To date, the functions of SAP11 and TCPs have been extensively studied, but the mechanism of the destabilization of TCPs triggered by SAP11 is still not clear. SAP11_PM19_ isolated from *Ca*. P. mali localizes in the cytoplasm and nucleus [4]. Based on this finding, the question arises as to what function SAP11_PM19_ might have in the cytoplasm and nucleus. By comparing phenotypes of transgenic Arabidopsis plant lines expressing the same amount of the *SAP11_PM19_* transgene, the transgenic plant lines expressing the protein exclusively in the nucleus (bi-NLS-SAP11_PM19_) show no crinkled leaf like the transgenic Arabidopsis plant lines expressing the protein exclusively in the cytoplasm (NES-SAP11_PM19_) and wt (SAP11_PM19_), while the number of stems is not significantly different. We demonstrated that SAP11_PM19_ can destabilize different AtTCPs in both the cytoplasm and nucleus (Figure 3). Moreover, AtTCP3 is one of the most important AtTCPs for the development of leaf morphology, which are commonly found in the cytoplasm and nucleus and are highly expressed in leaves [25]. Destabilization of this AtTCP in the cytoplasm could be very effective in reducing the amount of the TCP protein transported into the nucleus to control the transcription of target genes. In addition, one TCP can dimerize with another TCP to be transcriptionally active [25], so destabilization of the TCP in the cytoplasm could also reduce the amount of the TCP available for dimerization, resulting in lower activity of the TCP at the target site. Therefore, the presence of SAP11_PM19_ in the cytoplasm may be more effective in binding the TCP and causing destabilization of the TCP at the TCP translation site, whereas SAP11_PM19_ in the nucleus must compete for binding the TCP (and possibly induces destabilization of TCP) before the TCP binds the DNA target. AtTCP12 is one of the major AtTCPs controlling axillary meristem development [26,27,28] and is expressed at very low levels in plant cells [25]; therefore, triggering the destabilization of AtTCP12 by SAP11_PM19_, which is localized either in the cytoplasm or in the nucleus, might be sufficient to inhibit its functionality. Thus, this could be one of the explanation for the disappearance of the crinkled leaf phenotype and the remaining of the witches’ broom phenotype in the transgenic Arabidopsis plants expressing SAP11_PM19_ exclusively in the nucleus. Nevertheless, the presence of SAP11_PM19_ in the cytoplasm and nucleus could have a synergistic effect in strongly controlling TCP activity.

Strohmayer and co-workers [4] showed that SAP11_PM19_ can bind AtTCPs, especially of class I, but the binding does not always cause destabilization of the bound protein (AtTCP19). It was shown that DELLAs proteins (Arabidopsis nuclear proteins) modulate the plant development by blocking the DNA-binding domain of class I TCPs and thereby reduce their binding to their target promoters [29]. In contrast to the DELLAs proteins, our results show that SAP11_PM19_ binds to AtTCP19 in vitro, but it does not interfere with the DNA binding of AtTCP19 in an EMSA. However, the results of the in vivo protein–protein interaction analysis using the BiFC system show that SAP11_PM19_ binds AtTCP6 and relocalizes the protein from the nucleolus to the nucleoplasm, whereas it binds to AtTCP19 and relocalizes the protein to the cajal bodies in the nucleus. Hence, relocalization of AtTCP6 by SAP11_PM19_ could result in degradation of the protein by an unknown mechanism. Although no degradation was detected for AtTCP19 when co-expressed with SAP11_PM19_ [4], forming the cajal body complex with SAP11_PM19_ could nevertheless inhibit the functionality of AtTCP19. Some proteins relocalize TCPs in different manners. HopBB1 binds AtTCP14 and JAZ3 and relocalizes the complex, leading to protein degradation [11], whereas PRR2 translocates AtTCP19 or AtTC20 into cajal bodies and nuclear spackles, respectively, to stabilize the proteins [12]. Thus, the relocalizing of AtTCPs is a novel aspect of SAP11_PM19_ in addition to its destabilizing activity.

The function of SAP11 does not depend on plant species, but rather on TCP analogues and their function in each plant species [14]. It was shown that SAP11_MBSP_ could not bind and destabilized AtTCP3 and 4 while it bound and destabilized AtTCP12. Thus, the transgenic Arabidopsis plants expressing SAP11_MBSP_ showed no leaf phenotype, while the witches’ broom phenotype persisted [14]. From our results, only the witches’ broom phenotype is observed in all tested tobacco species expressing SAP11_PM19_, while only *N. occidentalis* shows the crinkled leaf phenotype as described for the transgenic Arabidopsis plants expressing this effector. Although the genetic data on TCPs in all tested tobacco species are not available, the result of genome wide analysis shows that in the genome of *N. tabacum* there is no TCP3 analogue to AtTCP3, but different types of TCP12 analogous to AtTCP12 are present [30]. The same finding was also shown in apple plant (malus domestic), host plant of ‘*Ca*. P. mali’ [31]. Thus, this could therefore be one of the possibilities to explain the loss of the crinkled leaf phenotype and the retention of the witches’ broom phenotype of transgenic *N. benthamiana* and *N. tabacum* expressing SAP11_PM19_ and in ‘*Ca*. P. mali’ infected apple. Our results thus confirm that the activity of SAP11 depends on the TCP analogues and their function in the individual plants [14]. Thus, using a model plant to study SAP11 activity should be carefully interpreted. 

Although the mechanism of destabilization of the SAP11-TCP complex is not yet known, it has been shown that proteasome inhibitors cannot inhibit the destabilization of AtTCPs by SAP11_AY-WB_ in a co-infiltrated experiment [32]. However, our results suggest that the destabilization can occur in the cytoplasm and nucleus; therefore, the inhibition of cytosolic proteasome by proteasome inhibitors may not sufficient to prevent the destabilization of AtTCPs by nuclear proteasome. It was shown that SWP1, a SAP11-like effector from wheat blue dwarf phytoplasma promotes the degradation of AtTCP18 (BRC1) via a proteasome system [5]. HopBB1 binds TCP and JAZ and the protein complex was destabilized by proteasome 26S [11]. In addition, it was shown that AtTCP17 was destabilized by the proteasome [33]. These results suggest that the proteasome system is involved in the control of TCPs in plant cells. Therefore, we cannot exclude the possibility that the proteasome may involve the destabilization of TCP in the presence of SAP11. From all the available data on SAP11–TCP interactions, we may deduce the function of SAP11 on TCPs as illustrated in Figure 8. In the cytoplasm, SAP11 binds to TCPs and triggers an unknown mechanism to degrade TCPs. The SAP11 and TCPs localize or are transported into the nucleus as single proteins or possibly as a SAP11–TCP complex. In the nucleus, SAP11 binds to TCPs and the TCPs are destabilized by an unknown mechanism in the nucleus. Furthermore, SAP11 can relocalize TCP(s), and the protein complex is destabilized (AtTCP6) or forms cajal bodies (AtTCP19) that are resistant to an unknown destabilizing mechanism, and the relocalized TCPs may thus be inactive. 

## 5. Conclusions

In conclusion, our results suggest that SAP11_PM19_ can cause the destabilization of AtTCPs both in the cytoplasm and nucleus. In addition, it can relocalize and possibly prevents some AtTCPs from becoming active at the target site (DNA or protein). Thus, the TCP relocating activity of SAP11_PM19_ becomes one of the functions of SAP11, which should be considered for the other SAP11s. Moreover, using a model plant to study the biochemistry of SAP11 on TCPs, the analogous features of TCPs in each plant species should be monitored.

## Figures and Tables

**Figure 1 microorganisms-10-01406-f001:**
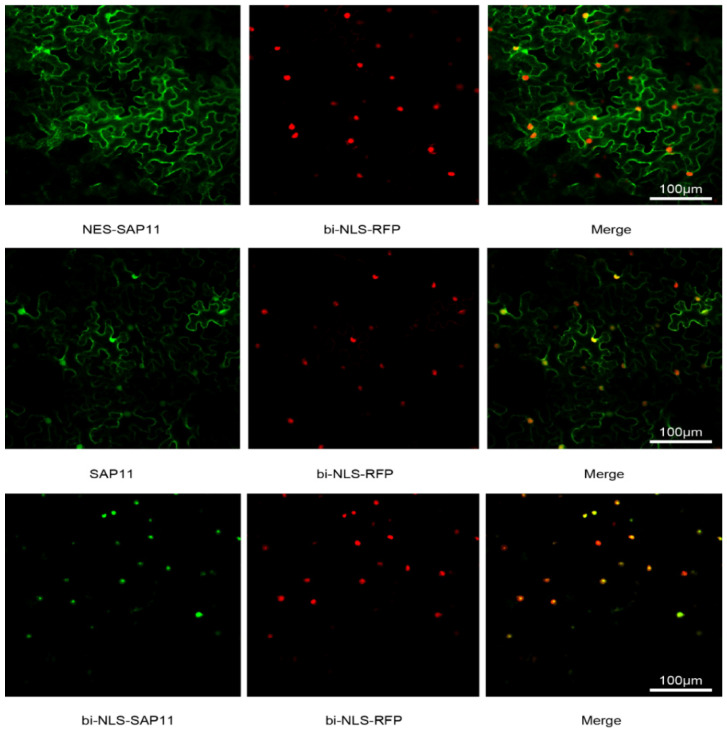
Transient expression of SAP11_PM19_, NES- and bi-NLS-SAP11_PM19_ in wt *N. benthamiana*. *SAP11_PM19_*, *NES*- and *bi-NLS-SAP11_PM19_* were fused to *GFP* and *bi-NLS* fused with *RFP* to mark the nucleus. The proteins were transiently expressed in *N. benthamiana* via *Agrobacterium*-infiltration. The localization of expressed proteins was analyzed by confocal microscopy using GFP and RFP filters. The result in Figure 1 shows that SAP11_PM19_-GFP (SAP11) localizes not only in the plant nucleus but also in the cytoplasm (middle panel) whereas NES (NES-SAP11, upper panel) and bi-NLS SAP11_PM19_-GFP (bi-NLS-SAP11, lower panel) were localized exclusively in the cytoplasm and nucleus, respectively. Co-localization with bi-NLS-RFP is indicated by yellow coloring in the merger.

**Figure 2 microorganisms-10-01406-f002:**
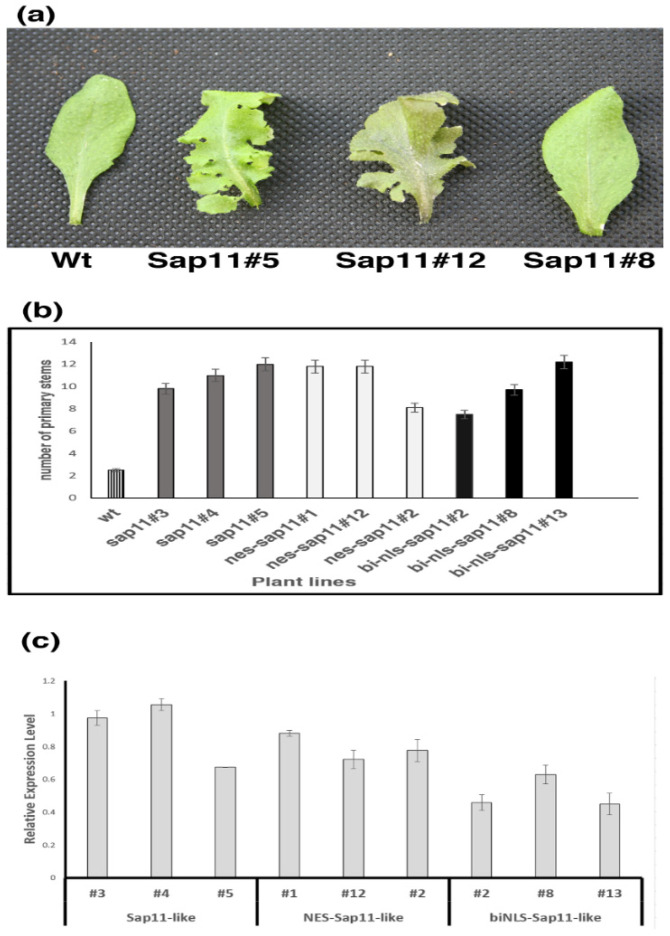
Transgenic plants expressing wt SAP11_PM19_ and its derivatives. Transgenic *A. thaliana* lines expressing *SAP11_PM19_* (Sap11), *NES-SAP11_PM19_* (NES-Sap11) and *bi-NLS-SAP11_PM19_* (NLS-Sap11) under the control of the *Cauliflower mosaic virus* 35S promoter compared to wt Arabidopsis Col-0 plant. All plants were grown for 8 weeks in long-day (16 h/8 h light/dark) conditions. For each of the three constructs, at least 3 lines were examined, all showing similar phenotypic characteristics. (**a**) Leaf morphology of transgenic and wt plant lines. (**b**) Number of shoots developed by transgenic *A. thaliana* lines compared to the wt Col-0 plant. (**c**) Relative expression levels of transgenes of transgenic lines compared to the geometric average of *glyceraldehyde-3-phosphate dehydrogenase* (*GAPDH*) and *protein phosphatase 2* (*PP2A*). All values show a *p*-value lower than 0.05 in ANOVA.

**Figure 3 microorganisms-10-01406-f003:**
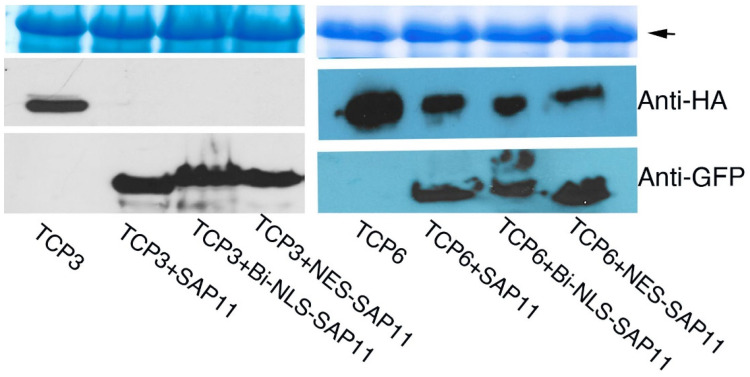
Western blot analysis of the destabilizing activity of SAP11_PM19_ in cytoplasm and nucleus. *AtTCP3-HA* (TCP3) and *AtTCP6-HA* (TCP6) were co-transiently expressed with *SAP11_PM19_-GFP* (SAP11), *NES* (NES-SAP11) or *bi-NLS-SAP11_PM19_-GFP* (Bi-NLS-SAP11) in *N. benthamiana* plants. The expressed proteins were extracted and subjected to an SDS-PAGE and Western blot analysis using anti-GFP for detecting SAP11_PM19_ expression (lower panel) and anti-HA-POD for detecting AtTCP3-HA and AtTCP6-HA (middle panel) expression. The Coomassie blue staining of a large subunit of Rubisco (arrow, upper panel) was used for controlling the total loading proteins.

**Figure 4 microorganisms-10-01406-f004:**
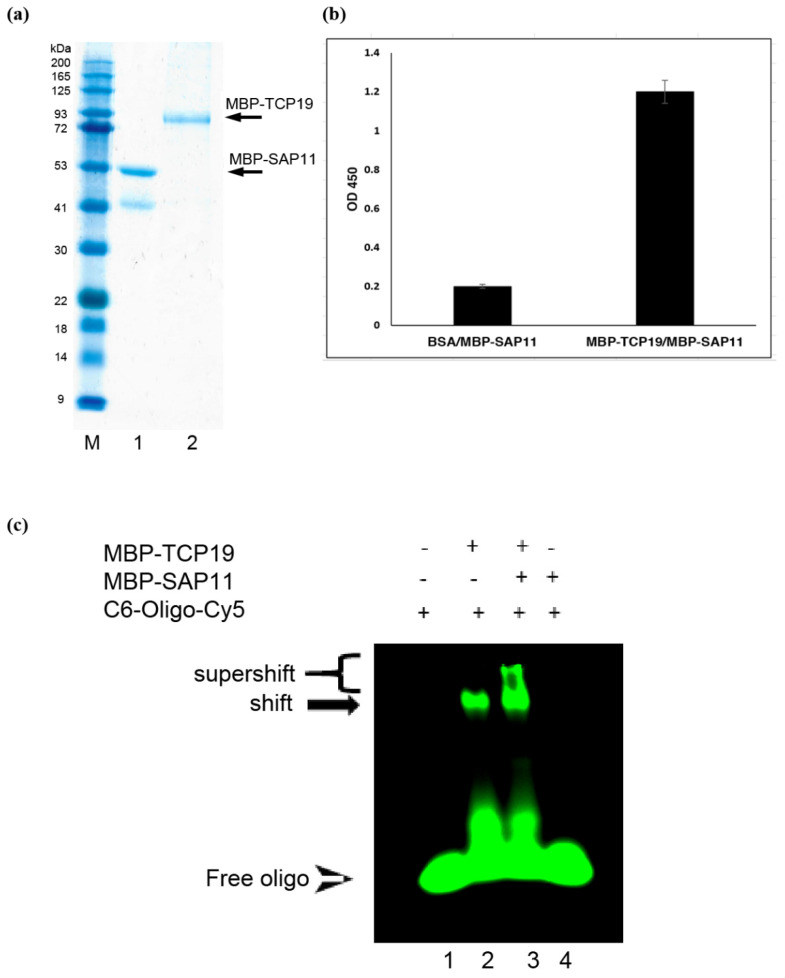
SAP11_PM19_ binds but does not inhibit DNA binding activity of AtTCP19 in vitro. (**a**) SDS-PAGE of the recombinantly expressed and purified proteins MBP-AtTCP19-His (MBP-TCP19) and MBP-SAP11_PM19_ (MBP-SAP11). M is a protein maker, lane 1 and 2 are MBP-SAP11_PM19_ and MBP-AtTCP19-His, respectively. (**b**) Recombinant MBP-SAP11_PM19_ binds to MBP-AtTCP19-His in an ELISA. The MBP-AtTCP19-His was coated on a microtiter plate and the MBP-SAP11_PM19_ was added. The binding activity was detected using anti-SAP11_PM19_ polyclonal antibody. BSA was used as a negative control. (**c**) EMSA agarose gel shows a band shift of C6-oligo-MBP-AtTCP19-His complex (lane 2). MBP-SAP11_PM19_ cannot hinder the binding of MBP-AtTCP19-His to the C6-oligo, but form a super-shift band (lane 3). In lane 1 and 4, BSA and MBP-SAP11_PM19_ cannot bind to C6-oligo-CY5 and are used as negative controls.

**Figure 5 microorganisms-10-01406-f005:**
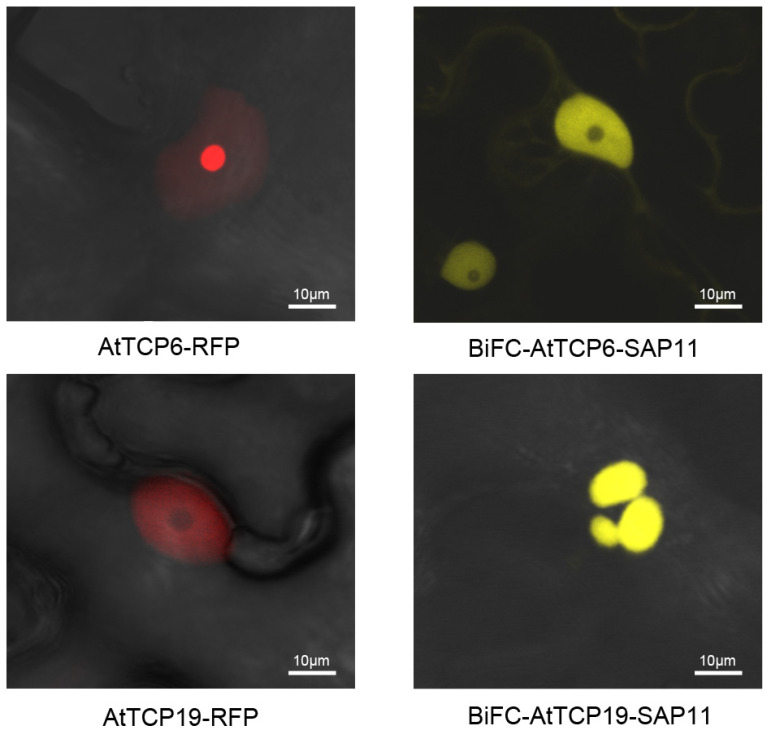
In vivo protein interaction of SAP11_PM19_ and AtTCP6 and 19. *SAP11_PM19_* and *AtTCP6* or *AtTCP19* were fused with each half of *YFP* in BiFC system. The proteins were transiently expressed via agroinfiltration in *N. benthamiana*. The protein interactions were visualized under a confocal microscope using RFP and GFP filters. The localizations of SAP11_PM19_ and AtTCP6 or AtTCP19 complexes (BiFC-AtTCP6/19-SAP11, **right** panel) are compared with the localization of AtTCP6 or AtTCP19 fused RFP (AtTCP6/19-RFP), respectively (**left** panel).

**Figure 6 microorganisms-10-01406-f006:**
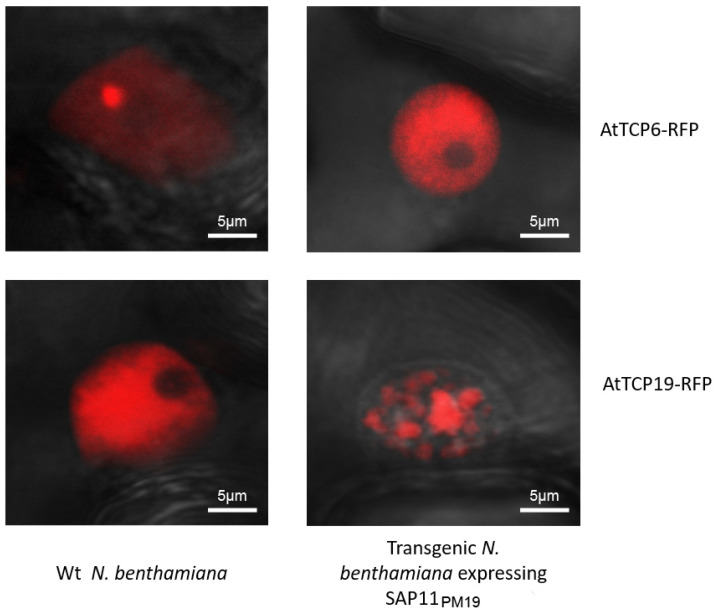
Transiently expressing AtTCP6 and 19-RFP in Wt and transgenic *N. benthamiana* plants expressing SAP11_PM19_*. AtTCP6 and 19* were fused with RFP and transiently expressed in Wt and in transgenic *N. benthamiama* plants expressing SAP11_PM19_. After 2 days, the expression and localization of the AtTCPs-RFP in infiltrated leaves were visualized under a confocal microscope.

**Figure 7 microorganisms-10-01406-f007:**
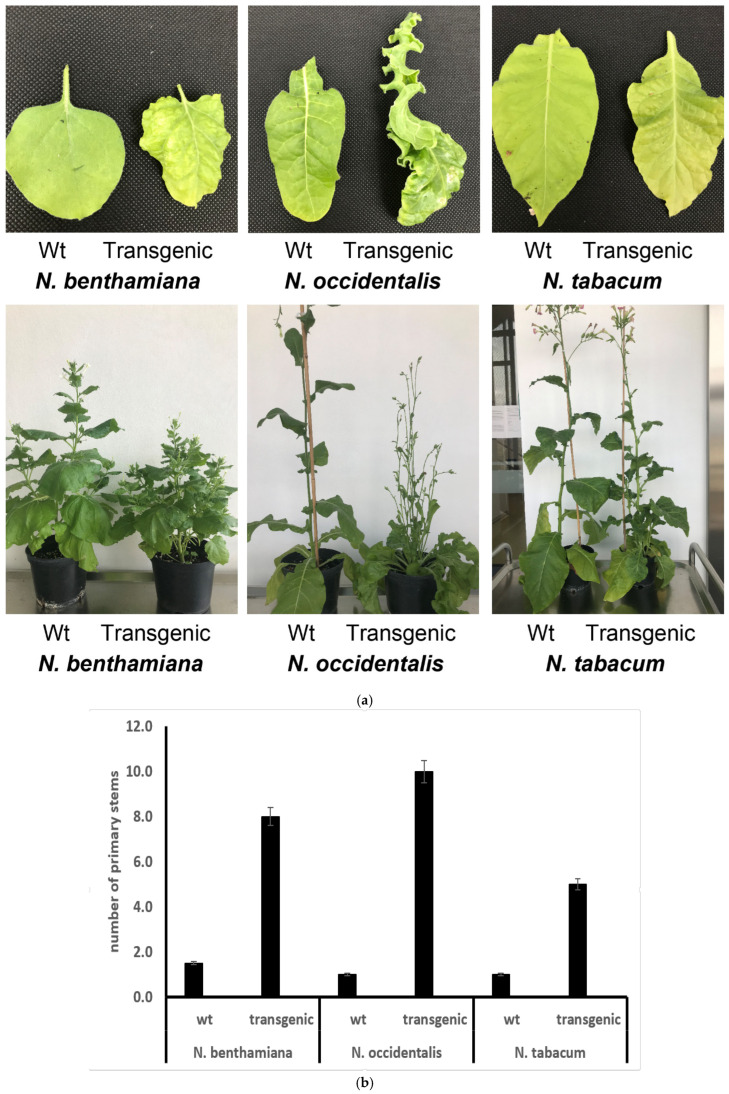
Transgenic tobacco plants expressing SAP11_PM19_. (**a**) Three different species of tobacco plant were used to establish transgenic plant lines expressing SAP11_PM19_ under the control of the *Cauliflower mosaic virus* 35S promoter. The phenotypes of three transgenic tobacco plant species were compared with wt. (**b**) Number of primary stems developed by 3 transgenic tobacco species compared to their corresponding wt.

**Figure 8 microorganisms-10-01406-f008:**
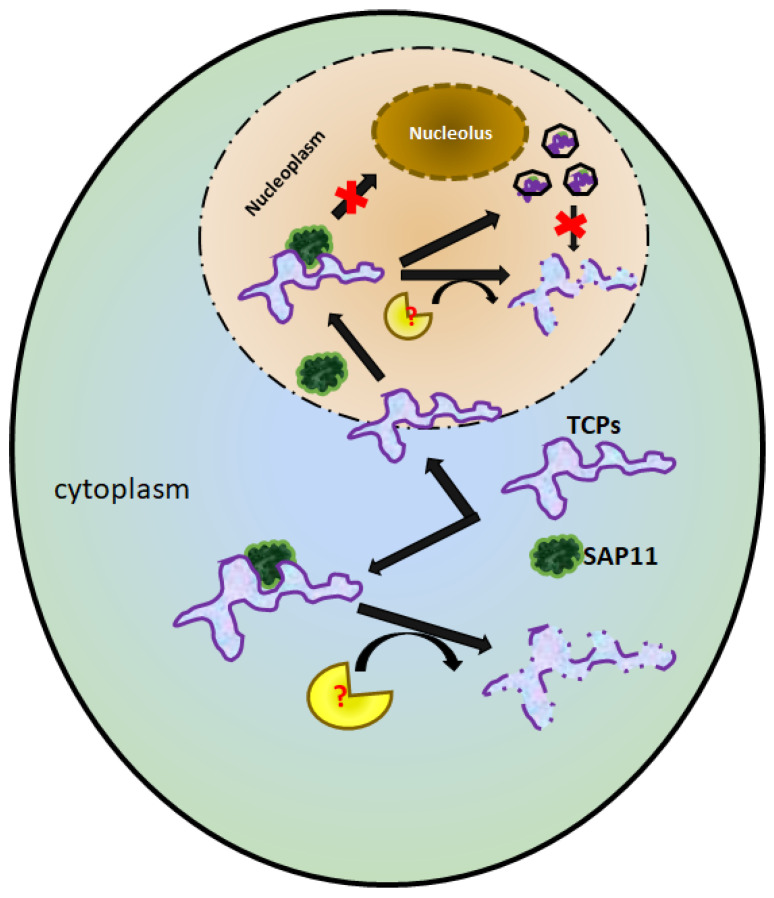
The activities of SAP11_PM19_ on AtTCPs. In the cytoplasm, the SAP11 (
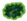
) binds TCPs (
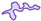
) and the TCPs are destabilized (
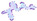
). The SAP11 and TCPs might separately migrate to the nucleus, and there SAP11 binds TCPs, resulting in either destabilization of the TCPs or relocalization of the TCPs, which can form cajal body complexes (
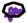
). 
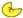
 is unknown TCP destabilizing machinery. 
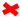
 is an inhibition.

## Data Availability

No new data were created or analyzed in this study. Data sharing is not applicable to this article.

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
