# Peer review of "Beyond Destabilizing Activity of SAP11-like Effector of *Candidatus* Phytoplasma mali Strain PM19"

_microorganisms, 2022, doi:10.3390/microorganisms10071406_

Round 1
Reviewer 1 Report
In this work, the authors show that the Phytoplasma SAP-like effector that causes witches’ broom and crinkled leaves in plants, destabilizes and relocalizes its target TCP. They discussed the association between the phenotype and the localization of effector. Several aspects of the study need further interpretation or additional experimentation.
1. In Introduction, the authors should introduce more background.
2. The images must be improved, for example, the author should provide the clear confocal images in Fig1 as Fig 5 and 6.
Author Response
Thank you very much for reviewing our Manuscript. Below you´ll find our responss to your suggestion and comments.
Reviewer 1
Comments and Suggestions for Authors
In this work, the authors show that the Phytoplasma SAP-like effector that causes witches’ broom and crinkled leaves in plants, destabilizes and relocalizes its target TCP. They discussed the association between the phenotype and the localization of effector. Several aspects of the study need further interpretation or additional experimentation.
- In Introduction, the authors should introduce more background.
We added more information in the introduction
- The images must be improved, for example, the author should provide the clear confocal images in Fig1 as Fig 5 and 6.
The resolution of pictures was improved. The quality of images was much better in the originally submitted files than in the files provided for review.
Reviewer 2 Report
Dear authors,
In your manuscript you investigated the effect ofPM19 (a SAP11 like effector of Phytoplasma mali) in transient and transgenic experiments. You found differences in the PM19 effector activity if it was expressed in the cytoplasm or in the nucleus. I found your manuscript very interesting, especially the model at the end. However, I found it also very difficult to follow. The main concept cannot be found, the interpretation of the results is not convincing. The materials and methods section miss important information. The results section contains information what would be better positioned into the introduction or to the discussion. I think that this manuscript should be reorganize and rewrite and English edited before resubmission to MDPI Microorganisms.
To help you for the revision I listed the main problematic points below.
main points
1/You cite long name of the investigated protein along the whole manuscript which makes it very difficult to read. I would suggest to identify an acronym for the investigated protein – PM19 would do. Make clear at the beginning that AP_SAP11-like_PM19 will be referred later as PM19 and use it later.
2/ During the whole manuscript you cite Strohmayer et al paper for several different aspect. This is good, but sometimes it would be easier to understand the whole story if you would include information from that paper for example:
2.1 if the PM19 origin was the vector and it was codon optimized for A Th, how reliable in could be expressed during agroinfiltration and transformation into N. benthamiana?
2.4 what species were transformed – you shoulddetail A Th, (which ecotype) and add information about tobacco transformation. Was floral dip was used in this later cases?
2.5 what reference gene was used as a reference. Did you used the same reference for nucleus expressing and cytoplasm expressing genes? This information is included later in the legend of Fig2, but two different genes are mentioned. Were they both used for the comparison? What their geometric average means?
3/ What was the promoter used for the wt form of the effector? line54 saiys that SMV signal was removed. What this means? It is not clear, please explain.
4/line 124-128 belongs to the materials and methods
5/ in 3.1 you should clearly write at the beginning that three constructs were prepared and tested first by transient, then in transgenic assay.
6/In the legend of Figure 1 please include the native wt version. In the title only the two with altered promoter is mentioned. Line139-147 I guess belongs to the legend, but in this current form it seems that it belongs tot eh main text. Please correct
7/ I would suggest to rearrange Figure 2. Legend below the leaves has strange characters, please correct – it could indicate the line from which they originate. I would put the column diagram about the number of primary stems below the picture, but starting with the wt column, and I would put the relative expression to the bottom of the figure. Again, I think that line169-178 belongs to the legend of the Figure. Please correct this.
8/Why TCP3 and TCP6 was used for co-expression experiments. You can add information about these specific TCPs in the introduction! Please add!
9/On Fig3 I can see that if PM19 was expressed TCP3 was disappeared for the system both in wt, and BI, or NES promoter-based version, while the level of TCP6 was not altered. So, the difference is based on the TCP and not on the different types of PM19. Please explain, if I am wrong. And again line 196-202 belongs to the legend.
10/ Please add information about SAP11 and TCP19 into the introduction. And include line 219-229 into the legend.
11/ line 230-234 belong to the introduction. TCP6 and TCP19 were selected for further experiments, but I am not convinced if we can be sure that they differ in their interaction with SAP11 as it is stated in line 237-238.
12/ On Figure 5 there are results from both TCP19 and TCP6 interaction, while in the title only the former one is mentioned. According to the pictures it is very difficult to see that they show the same result. Is it possible to show a picture about TCP6 and TCP19 localization without SAP11, to be able to see the changes?
13/ Figure 6 is about transient expression, but on the legend, it is stated that transgenic N. benthamiana was pictured, what is correct? It is not clear which construct were coinfiltrated in these plants? line 258-261 belongs to the legend.
14/ On Figure 7 you show the witched broom phenotype of the transformed tobacco plants, while the same information about A th were done on column diagram showing the number of stems. Can you add picture to the A th Figure and column diagram to the tobacco Figure? And again line 286-288 belongs to the legend.
15/ In the discussion line 352 I think that activity of SAP11 depends on the plant’s species, because they differ in the encoded TCP homologues, so there is a direct correlation!
16/ You discuss that there are no data about the TCPs encoded in tobacco, but you can extend the discussion with any possible data about TCPs in apple, the host of the investigated phytoplasma.
minor point
line40: What is AT and STAA stand for
line55: what SMV signal stand for?
line65:…ATHV strains and used for….
line262: please rephrase and correct “ different degree of phonotypes”
I think that after a very deep revision and English editing this manuscript can be reviewed again for the possibility to be accepted in MDPI Microorganisms.
Author Response
Thank you very much for reviewing our Manuscript. Below you´ll find our responss to your suggestion and comments.
Dear authors,
In your manuscript you investigated the effect of PM19 (a SAP11 like effector of Phytoplasma mali) in transient and transgenic experiments. You found differences in the PM19 effector activity if it was expressed in the cytoplasm or in the nucleus. I found your manuscript very interesting, especially the model at the end. However, I found it also very difficult to follow. The main concept cannot be found, the interpretation of the results is not convincing. The materials and methods section miss important information. The results section contains information what would be better positioned into the introduction or to the discussion. I think that this manuscript should be reorganize and rewrite and English edited before resubmission to MDPI Microorganisms.
To help you for the revision I listed the main problematic points below.
main points
1/You cite long name of the investigated protein along the whole manuscript which makes it very difficult to read. I would suggest to identify an acronym for the investigated protein – PM19 would do. Make clear at the beginning that AP_SAP11-like_PM19 will be referred later as PM19 and use it later.
We now use SAP11PM19 as abbreviation for AP_SAP11-like_PM19.
2/ During the whole manuscript you cite Strohmayer et al paper for several different aspect. This is good, but sometimes it would be easier to understand the whole story if you would include information from that paper for example:
We added some information from this paper to the introduction
2.1 if the PM19 origin was the vector and it was codon optimized for A Th, how reliable in could be expressed during agroinfiltration and transformation into N. benthamiana?
The codons were optimized for translation in planta, there should be no significant difference in A. thaliana and N. benthamiana in codon usage. Moreover the related figure shows a strong expression of codon optimized SAP11 in N. benthamiana, so it is clearly reliable.
2.4 what species were transformed – you should detail A Th, (which ecotype) and add information about tobacco transformation. Was floral dip was used in this later cases?
Done, information added
2.5 what reference gene was used as a reference. Did you used the same reference for nucleus expressing and cytoplasm expressing genes? This information is included later in the legend of Fig2, but two different genes are mentioned. Were they both used for the comparison? What their geometric average means?
The reference genes Glyceraldehyde-3-phosphate dehydrogenase (GAPDH) and protein phosphatase 2 (PP2A) were firstly mentionned in M&M and were used for nucleus and cytoplasm expression. Both of them were used for comparison. They were also used in our previous paper (Stromayer et al., 2021) and now were referenced in the text. The provided reference (Czechowski, T.; Stitt, M.; Altmann, T.; Udvardi, M.K.; Scheible, W.-R. Genome-wide identification and testing of superior reference genes for transcript normalization in Arabidopsis. Plant Physiol. 2005, 139, 5–17, doi:10.1104/pp.105.063743.) clearly shows, that traditional housekeeping genes can be outperformed in their stability of expression by other genes, amongst them different subunits of PP2A. At1g13320 used in this paper was one of the genes with very high stability especially in the development stage of A. thaliana.
3/ What was the promoter used for the wt form of the effector? line54 says that SMV signal was removed. What this means? It is not clear, please explain.
We do not know the original promotor of SAP11 in Ca. P. mali. But for plant expression in planta, 35s Promoter was used and now indicted in M&M. SVM encodes the sequence-variable mosaic protein signal sequence and the explanation for its removal was added to the respective paragraph of the manuscript.
4/line 124-128 belongs to the materials and methods
Done
5/ in 3.1 you should clearly write at the beginning that three constructs were prepared and tested first by transient, then in transgenic assay.
Done
6/In the legend of Figure 1 please include the native wt version. In the title only the two with altered promoter is mentioned. Line139-147 I guess belongs to the legend, but in this current form it seems that it belongs to the main text. Please correct
Done
7/ I would suggest to rearrange Figure 2. Legend below the leaves has strange characters, please correct – it could indicate the line from which they originate. I would put the column diagram about the number of primary stems below the picture, but starting with the wt column, and I would put the relative expression to the bottom of the figure. Again, I think that line169-178 belongs to the legend of the Figure. Please correct this.
Done
8/Why TCP3 and TCP6 was used for co-expression experiments. You can add information about these specific TCPs in the introduction! Please add!
Added
9/On Fig3 I can see that if PM19 was expressed TCP3 was disappeared for the system both in wt, and BI, or NES promoter-based version, while the level of TCP6 was not altered. So, the difference is based on the TCP and not on the different types of PM19. Please explain, if I am wrong. And again line 196-202 belongs to the legend.
In the respective figure, you can see that the signals of TCP3 and TCP 6 were reduced to different degrees when co-expressed with SAP in all constructs (SAP, NES and NLS-SAP11), compared to the expression of TCP3 and TCP6 without SAP. The destabilization of TCP6 is not as strong as TCP3 but one can clearly see the reducing of the signal for all constructs compared with the control (not with TCP3). This demonstrates that the altered expression level of TCP6 depends on the co-expression with SAP11PM19.
10/ Please add information about SAP11 and TCP19 into the introduction. And include line 219-229 into the legend.
Done
11/ line 230-234 belong to the introduction. TCP6 and TCP19 were selected for further experiments, but I am not convinced if we can be sure that they differ in their interaction with SAP11 as it is stated in line 237-238.
The line 230-234 were moved to the introduction. SAP11 can bind both TCPs but it can only destabilize TCP6. This result was shown by the given reference. You are right they do not differ in their binding behaviour but the effect of the interaction is different between the two TCPs. Not only the destabilizing activity of SAP11 on TCP6 and TCP19 is different, also the re-localization of TCP6 and 19 differ as shown in figure 5 and 6.
12/ On Figure 5 there are results from both TCP19 and TCP6 interaction, while in the title only the former one is mentioned. According to the pictures it is very difficult to see that they show the same result. Is it possible to show a picture about TCP6 and TCP19 localization without SAP11, to be able to see the changes?
TCP6 was added in the title. The localisation TCP6 and 19 when expressed alone were already shown in the figure, left panel.
13/ Figure 6 is about transient expression, but on the legend, it is stated that transgenic N. benthamiana was pictured, what is correct? It is not clear which construct were coinfiltrated in these plants?
As indicated in the text the transiently expressed TCP6 and TCP19-RFP on wt and transgenic plant expressing SAP11. So in the figure only infiltrated plants (wt, left panel and transgenic expressing SAP11 plant, right panel) are pictured.
line 258-261 belongs to the legend. Done
14/ On Figure 7 you show the witched broom phenotype of the transformed tobacco plants, while the same information about A th were done on column diagram showing the number of stems. Can you add picture to the A th Figure and column diagram to the tobacco Figure?
A column diagram indicating the number of primary stems in tobacco was added. A picture of Arabidopsis expressing SAP and the number of primary stems were already shown in figure 2. It is not necessary to show the same picture twice.
And again line 286-288 belongs to the legend. Done
15/ In the discussion line 352 I think that activity of SAP11 depends on the plant’s species, because they differ in the encoded TCP homologues, so there is a direct correlation!
Changed
16/ You discuss that there are no data about the TCPs encoded in tobacco, but you can extend the discussion with any possible data about TCPs in apple, the host of the investigated phytoplasma.
Added
minor point
line40: What is AT and STAA stand for
AT stands for “Apfeltriebsucht” which in German means “Appel proliferation”
STAA stands South Tyrol/Alto Adige
Added in the Text
line55: what SMV signal stand for?
SVM is sequence-variable mosaic protein signal sequence. Added in the text
line65:ATHV strains and used for….
Changed
line262: please rephrase and correct “different degree of phonotypes”
Done
I think that after a very deep revision and English editing this manuscript can be reviewed again for the possibility to be accepted in MDPI Microorganisms.